# Patterns of Drought Response of 38 WRKY Transcription Factors of *Zanthoxylum bungeanum* Maxim.

**DOI:** 10.3390/ijms20010068

**Published:** 2018-12-24

**Authors:** Xitong Fei, Lixiu Hou, Jingwei Shi, Tuxi Yang, Yulin Liu, Anzhi Wei

**Affiliations:** College of Forestry, Northwest Agriculture and Forestry University, Yangling 712100, China; feixt666@163.com (X.F.); houlixiu1994@163.com (L.H.); shijingweijw@163.com (J.S.); y2848@126.com (T.Y.)

**Keywords:** *Zanthoxylum bungeanum* maxim., transcription factor, WRKY, bioinformatics, abiotic stress

## Abstract

The WRKY family of transcription factors (TFs) includes a number of transcription-specific groupings that play important roles in plant growth and development and in plant responses to various stresses. To screen for WRKY transcription factors associated with drought stress in *Zanthoxylum bungeanum*, a total of 38 ZbWRKY were identified and these were then classified and identified with Arabidopsis WRKY. Using bioinformatics analyses based on the structural characteristics of the conservative domain, 38 WRKY transcription factors were identified and categorized into three groups: Groups I, II, and III. Of these, Group II can be divided into four subgroups: subgroups IIb, IIc, IId, and IIe. No ZbWRKY members of subgroup IIa were found in the sequencing data. In addition, 38 ZbWRKY were identified by real-time PCR to determine the behavior of this family of genes under drought stress. Twelve ZbWRKY transcription factors were found to be significantly upregulated under drought stress and these were identified by relative quantification. As predicted by the STRING website, the results show that the WRKYs are involved in four signaling pathways—the jasmonic acid (JA), the salicylic acid (SA), the mitogen-activated protein kinase (MAPK), and the ethylene signaling pathways. ZbWRKY33 is the most intense transcription factor in response to drought stress. We predict that WRKY33 binds directly to the ethylene synthesis precursor gene *ACS6*, to promote ethylene synthesis. Ethylene then binds to the ethylene activator release signal to activate a series of downstream genes for cold stress and osmotic responses. The roles of ZbWRKY transcription factors in drought stress rely on a regulatory network center on the JA signaling pathway.

## 1. Introduction

Transcription factors (TFs) are a class of proteins that specifically bind to *cis*-acting elements in the promoter region of eukaryotic genes. Through interaction between these and other proteins, the activation of the gene or the inhibition of transcription is achieved, thereby ensuring that the target gene is expressed at a specific intensity and at a specific time and place [1,2]. A wide variety of transcription factors are involved in the life processes of plants. They can be divided into many different types or families depending on the differences in sequence characteristics and functions. Families include WRKY, AP2/ERF, and MYB. Among these, WRKY is one of the largest in plants [3]. In 1994, the WRKY transcription factor SPF1 was discovered in sweet potato [4] and then cloned in *Arabidopsis thaliana* [5], rice [6], tomato [7], and cucumber [8].

The WRKY transcription factor has one or two WRKY domains containing the signature WRKYGQK peptide, which consists of approximately 60 highly conserved amino acid residues that bind to the W-box in the DNA sequence at the C-terminus of the WRKY domain. It also has a zinc finger structure (CX4-5CX22-23HX1H or CX7CX23HX1C), which is key in determining whether the WRKY protein binds to the related gene *cis*-acting element Wbox (TTGACT/C) [9,10]. According to the number of WRKY domains and the characteristics of zinc finger structures, WRKY proteins can be divided into three main types. Group I has two WRKY domains and the zinc finger structure type is C2H2. Groups II and III contain only one WRKY domain. In these, the zinc finger structure type of the group III members is C2HC, and of the group II members is C2H2. According to evolutionary relationships and certain amino acid motifs in the WRKY domain, group II WRKY transcription factors can be subdivided into five subgroups: subgroups IIa to IIe.

A large number of WRKY gene families play important roles in plant physiological processes such as growth and metabolism [11], response to various stresses [12,13], and defense responses to pathogens [14]. The WRKY family plays many key roles—especially in drought defense. *MtWRKY76* responds rapidly under drought stress, its overexpression increasing drought tolerance in *Medicago truncatula* [15]. VlWRKY48 was confirmed to increase catalase, peroxidase, and superoxide dismutase antioxidant enzymes under drought stress. This not only increases drought resistance in grapes but also plays an important role in resisting powdery mildew infection [16]. In *Arabidopsis*, WRKY46 has been shown to regulate responses to drought stress and is also involved in regulating light-dependent stomatal opening [17].

*Zanthoxylum bungeanum* (*ZB*), also known as Chinese prickly ash, belongs to the family Rutaceae and has a long history of cultivation in China. It is used both as a traditional medicine and as a food. Its fruit skin is one of the eight condiments [18], because of its unique numbing (anaesthetic) peppery taste. It has become an irreplaceable ingredient in traditional foods such as hot pot. In addition, *ZB* has a well-developed root system and excellent soil-fixing ability. Hence, it is also an ecological tree species for greening barren hills and conserving both water and soil. It plays an important role in the project of returning farmland to forest.

*Zanthoxylum bungeanum* is a drought-tolerant tree but, as an economic species, the development of the associated Chinese prickly ash industry can be seriously affected by geographical conditions and by variations in weather. Drought causes poor growth of the pepper and excessive fall of both flowers and fruits, and this has serious consequences for both yield and yield stability [19]. Therefore, increasing drought resistance and thus stabilizing the yield of *Zanthoxylum bungeanum* through genetic breeding will be important for ensuring the development of the Chinese prickly ash industry. Moreover, there are few studies on the transcription factors of *ZB* that seek to discover the corresponding ZbWRKY transcription factor under drought stress. These may provide valuable reference points for breeding drought-stress resistance into *ZB*.

## 2. Results

### 2.1. Identification of the WRKY Proteins in *Zanthoxylum bungeanum*

From the transcriptome of the *ZB* skin, 50 WRKY candidate genes were initially screened, based on the annotation information. The conserved domain of the WRKY gene was determined by the conservative domain prediction software Pfam to verify the correctness of the sequence alignment search. In the end, 38 sequences were identified as WRKY conserved domains. The sequence of ZbWRKY proteins was named according to the BLAST results of the Zanthoxylum protein sequence and the WRKY protein sequences of *Arabidopsis thaliana* (Table 1).

### 2.2. ZbWRKY Gene Phylogenetic Tree Analysis and Group Identification

The WRKY domain sequence of ZbWRKY protein was clustered using MEGA 7.0 software by analyzing the conserved domain characteristics of the WRKY proteins of *Arabidopsis thaliana*. The grouping of ZbWRKY proteins was determined using the type of known Arabidopsis and the citrus (*Citrus sinensis*) WRKY protein. The evolution of the ZbWRKY protein sequence was analyzed by MEGA 7.0. The ZbWRKY gene family was divided into Groups I, II, and III. Group II was further divided into four subgroups: subgroups II-b, II-c, II-d, and II-e (Figure 1 and Table 1). ZbWRKY41, ZbWRKY53, and ZbWRKY70 are difficult to cluster with other sequences, probably because of the large numbers of mutations in these sequences. From the protein sequence analysis, the ZbWRKY70 gene was mutated from the WRKYGQK to the WRKYGKK in the WRKY domain. ZbWRKY41 and ZbWRKY53 have a large number of mutation points outside the WRKY conserved region (Figure 2).

### 2.3. Conserved Domain Analysis of ZbKWRKY Gene

The WRKY domain of the ZbWRKY gene was analyzed by DNAMAN 8.0 sequence analysis software (Lynnon Biosoft, San Ramon, CA, USA) and online software WebLogo 3 (http://weblogo.berkeley.edu/logo.cgi). The results indicate that the WRKYGQK domain of the ZbWRKY gene is highly conserved. ZbWRKY members can be clearly divided into each subfamily, which is consistent with the clustering effect of the *Arabidopsis thaliana* and *Citrus sinensis* WRKY transcription factors, which can prove that the classification results of the ZBWRKY family are reliable (Figure 1).

### 2.4. ZbWRKY Gene Expression Analysis

The 38 ZbWRKY genes identified were used for relative quantitative analysis under drought stress. The results show that the 38 ZbWRKY genes had different effects on drought stress (Figure 3).

Some genes, including *WRKY33* and *WRKY15*, show high levels of expression in the early stages of drought stress. They also respond very rapidly. In contrast, *WRKY71* and *WRKY51* show significant responses only after 36 h of stress treatment. Among these, the expression levels of *ZbWRKY24(2)*, *ZbWRKY71*, *ZbWRKY70(2)*, *ZbWRKY31*, *ZbWRKY15*, *ZbWRKY33*, *ZbWRKY4*, *ZbWRKY75(2)*, *ZbWRKY70(3)*, *ZbWRKY4(2)*, *ZbWRKY1*, and *ZbWRKY51* showed significant upregulation. Generally, the expression levels of the 12 WRKY transcription factors respond significantly to drought, first increasing, then decreasing. This pattern of increase then decrease is consistent with that of gene expression change during drought stress. The relative expression levels of different *ZbWRKY* genes under different drought stress time were calculated with reference to the control group D0 (relative expression level defaulted to 1). The expression levels were more than 5-fold those at the D0 (control group) stage. Although the above genes are involved in obvious, corresponding processes for drought, the corresponding times and intensities are quite different. For example, ZbWRKY33 shows high expression at the beginning of the drought stress, more than 30-fold that at D0, but it then drops to the control level before rising again to a very high level after 48 h. The corresponding effect of ZbWRKY33 on drought is a phased response. There are many genes whose corresponding patterns are similar, but that differ in times and expression levels such as *ZbWRKY24(2)*, *ZbWRKY31*, and *ZbWRKY75(2)*. In addition, there are slow responding genes; expressions of *ZbWRKY70(2)* and *ZbWRKY71* remained low in the early stages of drought stress but later showed high levels of expression after treatment for 48 h.

## 3. Discussion

WRKY is one of the important transcription families in plant growth and development. It was first discovered in *Ipomoea batatas* [4] and has subsequently been found in many plants, including *Arabidopsis thaliana* [5], rice [6], tomato [7], and cucumber [8]. The WRKY family functions by participating in other signaling pathways to activate the responding genes. The interaction between the ZbWRKY protein and other regulatory proteins was predicted by STRING (http://string-db.org/), a system that searches for interactions between known proteins and predicting proteins. This interaction involved both direct physical interactions between proteins and the correlation of indirect functions between proteins. The relative expression of the gene was detected by real-time PCR. The analysis showed that the relative expression of the 12 *ZbWRKY* genes was 5-fold higher than the control and there was a significant response relationship. The functional relationship between the 12 ZbWRKY proteins was predicted and analyzed using the STRING protein interaction database. The species selection was the model plant *Arabidopsis thaliana*.

The 12 transcription factors that responded to drought were input into the STRING website to predict protein interactions. From the protein functional connection network, it can be seen that the WRKY transcription factors have complex interactions with a variety of proteins. Among them, WRKY33, WRKY70, WRKY15, WRKY40, and WRKY25 interact frequently with other proteins, and WRKY33 interacts with WRKY70, WRKY15, WRKY40, WRKY25, MYB51, ERF6, ERF104, salt tolerance zinc finger (STZ), ACS6, MPK3, MPK4, and MKS1, suggesting that these proteins may participate in specific physiological responses and the growth of plants (Figure 4). The results indicated that many members of the ZbWRKY transcription factor family had different responses to drought. For example, the *WRKY33* transcription factor was clearly expressed in the early stage of stress, whereas the *WRKY71* transcription factor showed significant increases only in the later stages of drought stress. It is inferred that different members of the *ZbWRKY* transcription factor family play different roles in the overall drought-stress response and exert their various functions in a concerted manner to cope optimally with adverse environments. Thus, the ZbWRKYs and other regulatory substances form a self-defense system. In addition, from the aspect of gene expression, the relative expression of *ZbWRKY33* under drought stress was more than 30-fold higher than that of the control, indicating that these proteins play active roles in resisting drought in plants. It can be inferred that it participates in many signaling pathways for self-protection during the time that *ZB* is subjected to drought. Overexpression of *WRKY25* or *WRKY33* can greatly increase NaCl tolerance in Arabidopsis and increases the sensitivity to ABA [20]. In addition, WRKY33 has been shown to be an important transcription factor in pathogen defense, modulating the antagonistic relationship between defense pathways that mediate responses to *Pseudomonas syringae* and necrotrophic pathogens [21,22]. The ectopic expression of the *Gossypium hirsutum WRKY6* gene significantly increased the salt tolerance of *Arabidopsis thaliana*, whereas the silencing of the *GhWRKY6* gene increased the sensitivity to abiotic stress in cotton [13].

The MAPK signaling pathway, the JA signaling pathway, the SA signaling pathway, and the ethylene signal pathway are also important signal pathways involved in plant responses to adverse environments. Multiple protein interactions reveal that multiple signaling pathways, such as the MAPK signaling pathways, are involved in drought stress. MPK3 has been shown to function as a phosphorylated WRKY protein and is involved in the MAPK signaling pathway [23]. *WRKY25* and *WRKY33* transcription factors act downstream of MPK4-mediated signaling and contribute to plant resistance [24]. In the MPK4-WRKY33 double mutant, the inhibition of PAD3 expression further supports the view that WRKY33 is an effector of MPK4 [25]. MKS1 is a regulatory protein of plant defense responses. The coupling activation of MPK4 regulation may be facilitated by coupling a kinase to a specific WRKY transcription factor [25]. Mitogen-activated protein kinase (MAPK) cascade, MAPK kinase 3 (MKK3)-MAPK 6 (MPK6), is activated by JA in Arabidopsis and is capable of inducing downstream wounding stress response genes (Figure 5) [26]. In addition, studies have shown that the MKK2 pathway plays a positive role in increasing plant resistance under salt stress [27]. At the same time, external stress signals can also stimulate receptors on the cell membrane to activate the MAPK signaling pathway [28].

Co-expression analysis indicates that the ZbWRKY transcription factor family has interplaying effects on plant growth and developmental processes as well as being involved in biotic and abiotic stress responses and participating in signal transduction in multiple signaling pathways including ethylene, ABA, JA, and SA. Plant survival depends on the twin abilities of rapid perception and rapid response, and these are regulated primarily by the JA signaling pathway [29,30]. From analysis of the ZbWRKY expression, it also has very frequent interaction with the JA signaling pathway. For example, WRKY70 is an important node of JA- and SA-mediated plant resistance and acts as an activator of the SA-inducible gene and a suppressor of the JA-responsive gene to integrate these two signals that mediate the mutual resistance of plants [31]. Salt tolerance zinc finger (STZ) transcriptional repressors are involved in abiotic stress responses and can inhibit the stress response genes *DREB1A* and *LTI78*. In addition, STZ may also be involved in the early signal response of JA, regulate the expression of the JA biosynthesis gene *LOX3*, and control the expression of *TIFY10A*/*JAZ1*, which is a key repressor in the JA signaling cascade [32,33]. Moreover, MYC2 is a major positive regulator of JA gene expression and is capable of binding to the G-box region upstream of the JA gene to increase gene expression level [34]. The activation of the ABA/JA pathway also promotes plant drought tolerance by reducing transpiration, reducing stomatal opening, and inhibiting growth retardation factors [35]. These two signaling pathways have some common components in the guard cells; cADPR and cGMP are the second messenger in the ABA-induced stomatal closure signaling pathway. Studies have found that cADPR and cGMP also play the same role in JA-mediated stomatal closure [36].

The WRKY transcription factors are also closely related to the ethylene signaling pathway. ACS6 (1-aminocyclopropane-1-carboxylate synthase 6) is a synthetic precursor of ethylene, and WRKY33 has been shown to bind directly to the promoters of ACS2 and ACS6 to alter gene expression in Arabidopsis [37]. ERF6 (ethylene-responsive transcription factor 6) may be a transcriptional activator that binds to the GCC-box pathogenesis-related promoter element. It may be involved in the regulation of gene expression by components of stress factors and stress signal transduction pathways. Moreover, SA can induce the degradation of ERF transcription factor ORA59 and activate the JA signaling pathway at the same time. In addition, SA negatively regulates WRKY transcription factors and directly or indirectly inhibits JA response gene expression [38].

From the above analysis, there is a close relationship between STZ, MYB51, MPK3, MPK4, and ERF6. It can be inferred that external stresses stimulate the internal regulation system of organisms including WRKY, and also activate a large number of functional proteins and transcription factors involved in plant stress resistance. Among the many ZbWRKYs, ZbWRKY33 is especially worthy of attention. *WRKY33* responded significantly to the drought stress of *ZB*, and it was also demonstrated in other studies that it promoted plant drought resistance. Studies have shown that WRKY25, WRKY26, and WRKY33 regulate the cooperation between ethylene-activated protein- and heat shock protein-related signaling pathways, which mediate the response to heat stress. These three proteins interact functionally and play overlapping and synergistic roles in heat resistance in plants [39]. In addition, studies have shown that pathogens can induce the expression of *WRKY33* transcription factors, thereby activating the defense pathway and antagonizing the pathogens [21]. In general, WRKY33 participates in many biological activities and plays different roles with different protein interactions, forming a WRKY33-based anti-stress control system, and playing an especially prominent role in resisting external adversity.

Transcription factors respond to stress and participate in the regulation of numerous signaling pathways. They also produce protective substances that help to combat adverse environments. Previous studies have found that the transgenic wheat *TaWRKY93* gene *Arabidopsis thaliana* accumulates more SOD and POD than wild-type Arabidopsis This indicates that transgenic plants reduce cell membrane damage and increase intracellular antioxidant enzymes during drought [40]. Both drought stress and salt stress are osmotic stresses. These two stresses produce many similar results, such as stomatal closure, increased antioxidants, and increased water retention. Based on the important role of the WRKY family in drought resistance, disease resistance, and other roles, it is important to consider overexpression of certain members of the resistant WRKY family when seeking to improve the resistance characteristics of plants.

## 4. Materials and Methods

### 4.1. Materials

Three-month-old Chinese prickly ash seedlings (Fengxian *Zanthoxylum bungeanum* test station, Baoji, China) were selected for uniformity. The seedlings were cultivated in the environmental-controlled greenhouses of Northwest Agriculture and Forestry University (Xianyang, China). The culture conditions were a temperature of 25 ± 3 °C and a light intensity of 2000 lx, 12 h light/day. Half-strength Murashige and Skoog (MS) medium with additions of polyethylene glycol (PEG) 6000 to 20% was used as the drought-stress treatment. Seven treatment times were employed: 0 h (D0), 3 h (D3), 6 h (D6), 12 h (D12), 24 h (D24), 36 h (D36), and 48 h (D48). The leaves were collected and each treatment combination was replicated three times. The samples collected were promptly stored in liquid nitrogen pending analysis. Samples of the compounds are available from the authors.

Transcriptome sequencing materials of *Zanthoxylum bungeanum* were collected from the Experimental Station of Northwest Agriculture and Forestry University in Fengxian, Shaanxi, China. The *Zb* skins were collected and quickly immersed in liquid nitrogen and stored at −80 °C pending use for transcriptome sequencing. From the transcriptome of the *Zb* skin (unpublished), the ZbWRKY transcription factor was initially obtained based on the annotation information. The WRKY transcription factor was confirmed by the SMART website (http://smart.embl-heidelberg.de/smart/set_mode.cgi?NORMAL=1) and the HMMER website (https://www.ebi.ac.uk/Tools/hmmer/) and the sequence of 38 *Zb* WRKY transcription factors was then obtained.

### 4.2. Methods

#### 4.2.1. Total RNA Extraction and Reverse Transcription

The total RNA of the sample is extracted using the TaKaRa MiniBEST Plant RNA Extraction Kit (TaKaRa, Beijing, China) and extracted and purified according to the instructions. The obtained sample total RNA was tested for its concentration and purity using NanoDrop 20000 (Thermo Scientific, Pittsburgh, PA, USA). Only RNA samples with OD_260/280_ ratios between 1.8 and 2.2 and OD_260/230_ ratios higher than 2.0 were used for subsequent experiments.

#### 4.2.2. Bioinformatics Analysis

MEGA 7.0 (Center for Evolutionary Medicine and Informatics, Tempe, AZ, USA) was used in the phylogenetic tree analysis of the WRKY protein sequences of the *ZB* and the WRKY protein sequences of *Arabidopsis thaliana* [41]. The set parameters were as follows: Poisson correction, pairwise deletion, and bootstrap (1000 repetitions). Meanwhile, 35 Arabidopsis WRKY protein sequences were introduced to subclass *ZbWRKY*. The Arabidopsis WRKY protein sequences were downloaded from the Arabidopsis website (http://www.arabidopsis.org/), and *Citrus sinensis* WRKY protein sequences were downloaded from the PlantTFDB website (http://planttfdb.cbi.pku.edu.cn/). The analysis of the heat map was carried out using the OmicShare website (http://www.omicshare.com/).

#### 4.2.3. Identification and Screening of Transcription Factor Family Genes

According to the annotation information of transcriptome data, ZbWRKY transcription factors were preliminarily screened out, then BLAST confirmation was carried out according to the conservative structural domain of WRKY, and 38 sequences satisfying the requirements were finally confirmed from 50 transcription factors.

#### 4.2.4. Real-Time qPCR

All primers were designed using Primer 5.0 (Premier, Palo Alto, CA, USA) and the primers for the sequences are listed in Table 2. To obtain the expression pattern of the WRKY family gene under drought stress, CFX96 Real-Time PCR Detection System (Bio-Rad, Hercules, CA, USA) was used to detect gene expression levels. Each 10 μL reaction used 5 μL of 2× SYBR Premix Ex Taq II (TaKaRa, Beijing, China), 1 μL of cDNA, 1 μL of each of the forward and reverse primers and 3 μL of ddH_2_O. RT-qPCR amplifications were carried out using the program: 95 °C for 30 s followed by 40 cycles of 94 °C for 5 s, 54 °C for 30 s, and 72 °C for 45 s. To accurately measure the relative expression level of genes, 1 ng of cDNA was added to each fluorescent quantitative PCR reaction system.

*ZbUBQ* was used as a reference gene to calculate the relative expressions of the WRKY family genes [42]. The cycle threshold values (Ct) of all samples were collected by the CFX96 Real-Time PCR Detection System, and gene relative expression was calculated as follows: relative expression, that is, 2^−ΔΔ*C*t^.

## 5. Conclusions

Thirty-eight WRKY transcription factors were identified and isolated from *ZB*. WRKY transcription factors were divided into three groups (Groups I, II, and III) according to the conserved sequence of transcription factors and the family classification of *Arabidopsis thaliana*—of the 38 WRKY transcription factors, 15 were in Group I, 16 in Group II, and 7 in Group III.

The relative expression levels of 38 transcription factors under drought stress induced by PEG 6000 were detected using real-time PCR. All transcription factors responded differently to drought stress, and 12 ZbWRKY transcription factors were significantly up-regulated. These results suggest that these transcription factors may be involved in the drought-resistant process of *ZB*. Protein interaction analysis of the responding genes was carried out on the STRING website. *Arabidopsis thaliana* was selected as the predicted species. The results of the interaction analysis indicate that the WRKY transcription factor family is widely involved in the transmission of plant stress signals and is closely related to the JA signaling pathway, the MAPK signaling pathway, the ethylene signaling pathway, and the ABA signaling pathway. *ZbWRKY33* is a homologous sequence of *Arabidopsis thaliana WRKY33*, which is the most intense transcription factor in response to drought stress (the relative expression level is more than 30-fold higher than that of the control group). WRKY33 is able to bind directly to the ethylene synthesis precursor gene *ACS6*, to promote ethylene synthesis. Subsequently, ethylene binds to the ethylene-activated protein release signal to activate a series of downstream cold-stress and osmotic-stress response genes, suggesting that *ZbWRKY33* has a similar function in *ZB*. The JA signaling pathway plays an important role in plant stress signaling. The role of ZbWRKY transcription factors in drought stress relies on a regulatory network center in the JA signaling pathway. Based on the known functions of transcription factors such as *ZbWRKY33*, these transcription factors can be used as candidate genes for drought-resistance breeding.

## Figures and Tables

**Figure 1 ijms-20-00068-f001:**
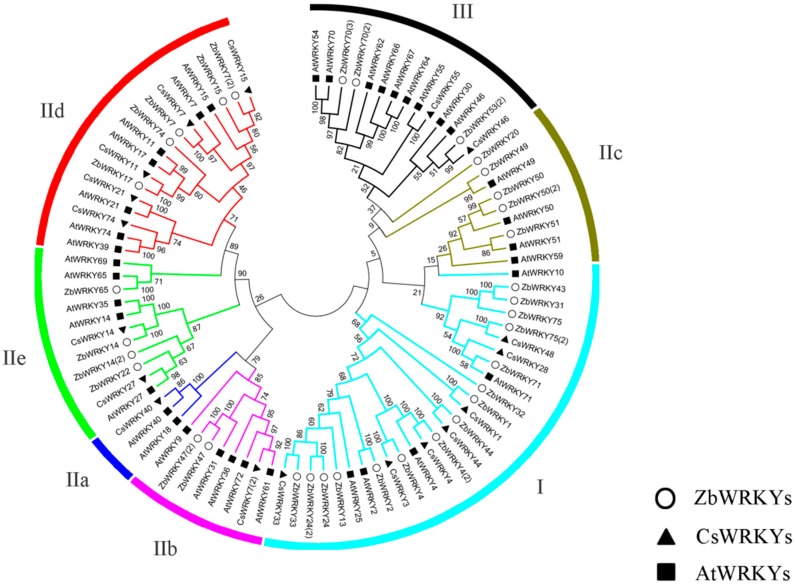
Phylogenetic tree of the WRKY protein.

**Figure 2 ijms-20-00068-f002:**
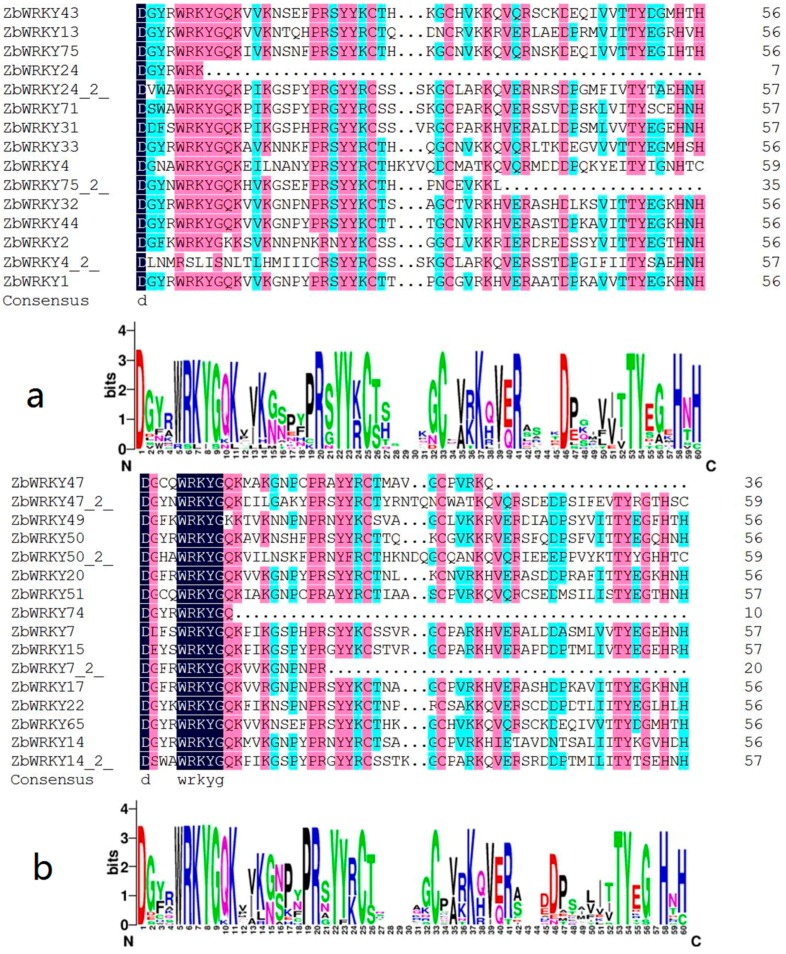
Multiple alignment of WRKY proteins in *Zanthoxylum bungeanum*: (**a**) Group I, (**b**) Group II, and (**c**) Group III.

**Figure 3 ijms-20-00068-f003:**
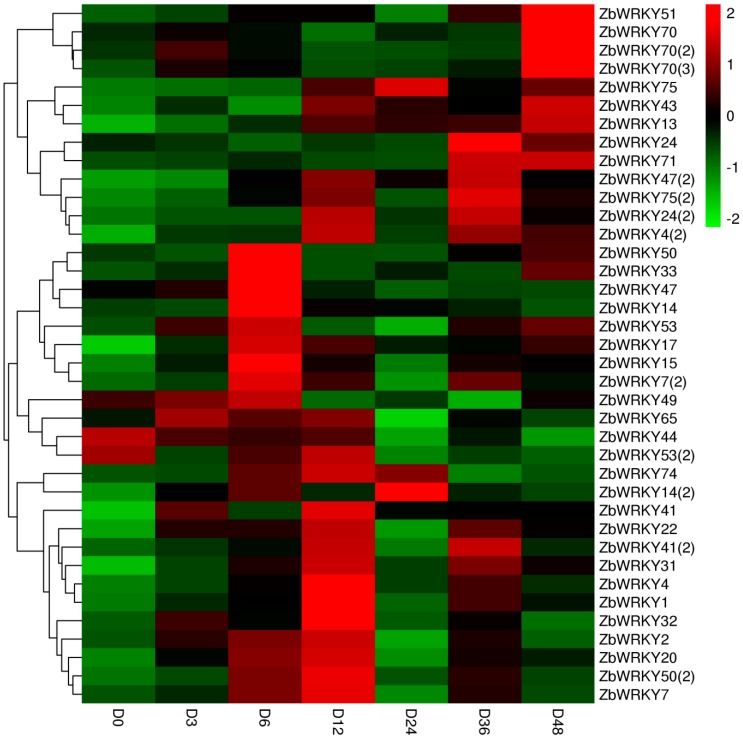
Relative expression heat map of the 38 ZbWRKY genes under drought stress.

**Figure 4 ijms-20-00068-f004:**
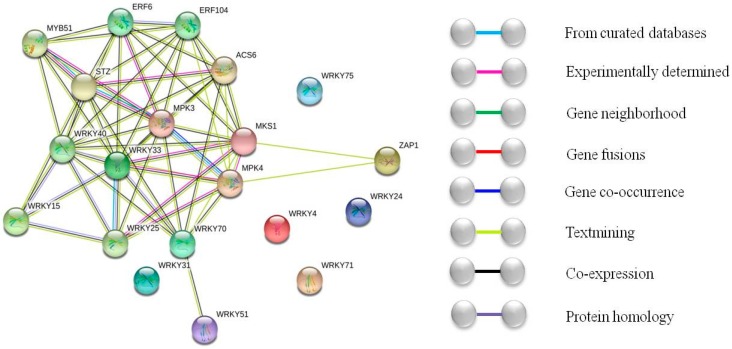
The WRKY protein functional connection network.

**Figure 5 ijms-20-00068-f005:**
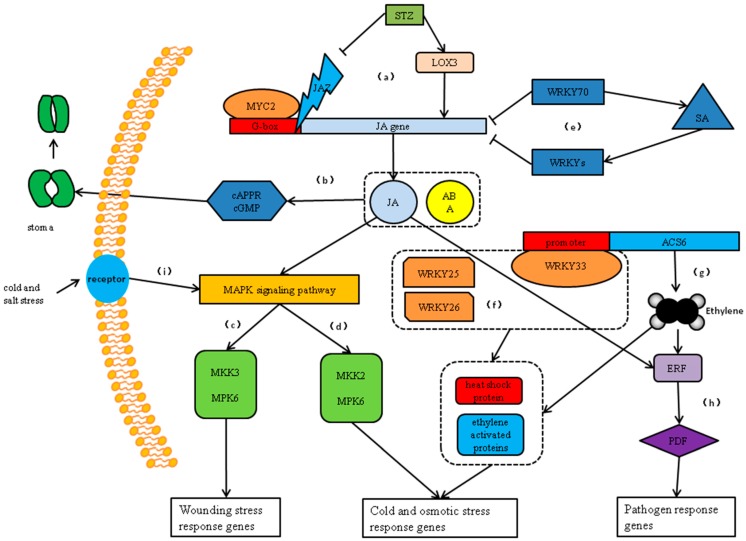
The stress signal path model of WRKY participation. (**a**) MYC2 is a major positive regulator of JA gene expression and is capable of binding to the G-box region upstream of the JA gene to increase gene expression levels. JAZ can suppress the combination of MYC2 and G-box area, and salt tolerance zinc finger (STZ) is involved in the early signaling response of JA and regulates the expression of the JA synthetic gene *LOX3*. (**b**) The JA/ABA signaling pathway regulates the second messenger cAPPR and cGMP to close stomata. (**c**) The mitogen-activated protein kinase (MAPK) cascade, MAPK kinase 3 (MKK3)-MAPK 6 (MPK6), is activated by JA in Arabidopsis and is capable of inducing downstream wounding stress response genes. (**d**) MPK6 activation by MEKK1 is mediated by MKK2. MKK2–MPK6 cascades mediate cold and salt stress tolerance in plants. (**e**) WRKY70 can inhibit the expression of the *JA* gene and activate the SA signaling pathway to induce more WRKY transcription factors to inhibit JA. (**f**) WRKY25, WRKY26, and WRKY33 transcription factors interact to regulate cooperation between ethylene-activated protein- and heat shock protein-related signaling pathways. (**g**) ACS6 is a synthetic precursor of ethylene, and WRKY33 binds directly to the promoters of ACS2 and ACS6 to promote gene expression. (**h**) ERF can be activated by ethylene and act on PDF (plant defensin) to activate downstream pathogen response genes, which can also be regulated by JA. (**i**) When the receptor receives signals of cold or osmotic stress, it will stimulate the MAPK signal pathway and thus improve plant resistance.

**Table 1 ijms-20-00068-t001:** *Zanthoxylum bungeanum* WRKY protein sequence classification and domain.

Number	Name	Group	Structural Domain	Homologous Gene of *Arabidopsis thaliana*
1	*ZbWRKY43*	I	WRKYGQK + WRKYGQK + C2H2	KDO64417.1
2	*ZbWRKY13*	I	WRKYGQK + WRKYGQK + C2H2	XP_006464513.1
3	*ZbWRKY47*	II-b	WRKYGQK + C2H2	XP_006492428.1
4	*ZbWRKY75*	I	WRKYGQK + WRKYGQK + C2H2	XP_006436871.1
5	*ZbWRKY41*	III	WRKYGQK + C2HC	XP_011022522.1
6	*ZbWRKY74*	II-d	WRKYGQK + C2H2	KDO54401.1
7	*ZbWRKY24*	I	WRKYGQK + WRKYGQK + C2H2	XP_006431962.1
8	*ZbWRKY24(2)*	I	WRKYGQK + WRKYGQK + C2H2	XP_006431962.1
9	*ZbWRKY22*	II-e	WRKYGQK + C2H2	XP_006444950.1
10	*ZbWRKY70*	III	WRKYGKK mutation	XP_006435943.1
11	*ZbWRKY49*	II-c	WRKYGQK + C2H2	XP_006480891.1
12	*ZbWRKY50*	II-c	WRKYGQK + C2H2	XP_006474455.1
13	*ZbWRKY50(2)*	II-c	WRKYGQK + C2H2	XP_006474454.1
14	*ZbWRKY71*	I	WRKYGQK + WRKYGQK + C2H2	KDO78100.1
15	*ZbWRKY70(2)*	III	WRKYQGQK + C2HC	XP_006481203.1
16	*ZbWRKY65*	II-e	WRKYGQK + C2H2	XP_006467614.1
17	*ZbWRKY7*	II-d	WRKYGQK + C2H2	KDO84087.1
18	*ZbWRKY31*	I	WRKYGQK + WRKYGQK + C2H2	KDO54715.1
19	*ZbWRKY15*	II-d	WRKYGQK + C2H2	XP_006425967.1
20	*ZbWRKY7(2)*	II-d	WRKYGQK + C2H2	XP_006494483.1
21	*ZbWRKY17*	II-d	WRKYGQK + C2H2	XP_006483548.1
22	*ZbWRKY33*	I	WRKYGQK + WRKYGQK + C2H2	XP_012089749.1
23	*ZbWRKY4*	I	WRKYGQK + WRKYGQK + C2H2	XP_006482724.1
24	*ZbWRKY53*	III	WRKYQGQK + C2HC	XP_006491244.1
25	*ZbWRKY75(2)*	I	WRKYGQK + WRKYGQK + C2H2	AEO31515.1
26	*ZbWRKY70(3)*	III	WRKYQGQK + C2HC	KDO67551.1
27	*ZbWRKY14*	II-e	WRKYGQK + C2H2	XP_006448817.1
28	*ZbWRKY20*	II-c	WRKYGQK + C2H2	XP_006452514.1
29	*ZbWRKY47(2)*	II-b	WRKYGQK + C2H2	XP_006444621.1
30	*ZbWRKY32*	I	WRKYGQK + WRKYGQK + C2H2	XP_006453624.1
31	*ZbWRKY44*	I	WRKYGQK + WRKYGQK + C2H2	XP_006428919.1
32	*ZbWRKY53(2)*	III	WRKYQGQK + C2HC	XP_006439376.1
33	*ZbWRKY2*	I	WRKYGQK + WRKYGQK + C2H2	XP_006474751.1
34	*ZbWRKY4(2)*	I	WRKYGQK + WRKYGQK + C2H2	KDO83046.1
35	*ZbWRKY1*	I	WRKYGQK + WRKYGQK + C2H2	XP_006447745.1
36	*ZbWRKY41(2)*	III	WRKYQGQK + C2HC	XP_006490540.1
37	*ZbWRKY51*	II-c	WRKYGQK + C2H2	XP_006464492.1
38	*ZbWRKY14(2)*	II-e	WRKYGQK + C2H2	KDO76342.1

**Table 2 ijms-20-00068-t002:** Primers for real-time quantitative PCR.

Number	Name	F/R	Primer Sequences	Tm	Product Length	Homologous Sequence of *Arabidopsis thaliana*
1	*ZbWRKY43*	F	TTCAGCTTCCCAGTTCTCATT	56.6	144	KDO64417.1
		R	TGTTTCACCTTATCATTACTCCC	55.8		
2	*ZbWRKY13*	F	TTTTCAGCTTCCCAGTTCTCA	57.6	146	XP_006464513.1
		R	TGTTTCACCTTATCATTACTCCC	55.8		
3	*ZbWRKY47*	F	AGAAGCGAGTAACCTTAAAGTAGA	54.7	203	XP_006492428.1
		R	TGACATTCAATGCAGCAGAT	54.3		
4	*ZbWRKY75*	F	GATGATGGTTATAGATGGAGGA	53.5	202	XP_006436871.1
		R	GTTCAAAGCTGTCAGTGAGTTT	54		
5	*ZbWRKY41*	F	GAGCAATCTGGACTTTGAGGGA	60.8	186	XP_011022522.1
		R	CCAAGAATGTCTTTTTGACCGT	58.7		
6	*ZbWRKY74*	F	CAAGCACTTTGTCATCAACTAA	53.5	158	KDO54401.1
		R	ACTCCCAGAATCTTCTCCTCTG	56.8		
7	*ZbWRKY24*	F	TCATCCTAAGCCTCAATCTACC	55.7	212	XP_006431962.1
		R	ATTTACTCCTCTGTGATCCCTG	55.3		
8	*ZbWRKY24(2)*	F	CAACAAAGAAGAAAGTGGAGAGG	57.9	135	XP_006431962.1
		R	GGATGGCATTAGAATTAACCGAA	60.5		
9	*ZbWRKY22*	F	GAAACAAGTGGAGCGAAACAGA	59.6	206	XP_006444950.1
		R	CCGGAGATGAGGAAGAGAAAGT	59.3		
10	*ZbWRKY70*	F	TGAAGATGGTCATGGATGGAGA	59.9	62	XP_006435943.1
		R	TTGGATATTTAGCATTGCGGAT	59.5		
11	*ZbWRKY49*	F	GGCTTCATCTCCACTTTGCTTAC	60.1	87	XP_006480891.1
		R	TGGTCTTTTTTGGTTTCTTTGTT	58.1		
12	*ZbWRKY50*	F	CATTTTCGGGGCTGTTCATA	58.6	122	XP_006474455.1
		R	CCTCTCCCTCCCACTTTCAT	57.9		
13	*ZbWRKY50(2)*	F	TATGGGAAGAAGACGGTGAA	55.4	156	XP_006474454.1
		R	ATTGCTTTGGTGAGTGTGGA	55.9		
14	*ZbWRKY71*	F	GAAGATGGGTATCGGTGGA	55	189	KDO78100.1
		R	CAAGGTTGTGGGAAGTGGA	56		
15	*ZbWRKY70(2)*	F	AAATCTCATTCATAAGTCCAC	56.4	262	XP_006481203.1
		R	GCTTCTGATAATCTTCCCAAC	53.4		
16	*ZbWRKY65*	F	TTCAAAAGCCGCCAAGACC	61	153	XP_006467614.1
		R	TCCATTTCCCCAAACCACG	61.1		
17	*ZbWRKY7*	F	TGCCTCCTTTGCCTCTCCA	61.2	147	KDO84087.1
		R	CGCTGTCAGTATCCCCTGC	58.6		
18	*ZbWRKY31*	F	AAAGGAAATGGTAAACGAGG	54	222	KDO54715.1
		R	TCTGAAGAAGAAAGCGAAGG	55		
19	*ZbWRKY15*	F	GATTGTCTGACATCCCACC	52	147	XP_006425967.1
		R	CATCCAAAGCTCTCTCCAC	52.2		
20	*ZbWRKY7(2)*	F	GGAGAAAGTACGGACAGAAACC	57.4	136	XP_006494483.1
		R	GTGACAACAAGCATTGATGGAT	57.4		
21	*ZbWRKY17*	F	AACGGGAAGCAAGGAGGAT	58.5	91	XP_006483548.1
		R	TGTACGGCTGAGAGGCGAG	59.4		
22	*ZbWRKY33*	F	CTTCCTCCTGTTCCCCCTTCT	61.2	257	XP_012089749.1
		R	CGTGGTTCCTGTTGCCTCTTA	60.3		
23	*ZbWRKY4*	F	ACTTGTTTTCCATCCGTCAC	54.6	158	XP_006482724.1
		R	CCATCATCAGCAGGTTTGTC	55.8		
24	*ZbWRKY53*	F	TCCTTTCCCAACACTACACCAA	59.6	116	XP_006491244.1
		R	AGAGTCCAGCCGACCTTTACAT	59.7		
25	*ZbWRKY75(2)*	F	ATAATAAGTGGGGTTCTTTTGGTT	58.1	132	AEO31515.1
		R	GTTGTTGTTGATGTTGTAGTGGTG	57.4		
26	*ZbWRKY70(3)*	F	TCCCAACAGAAGTGAAACAAG	55.2	95	KDO67551.1
		R	CACAAAATCCTCCCACATAAT	54.3		
27	*ZbWRKY14*	F	CCATGTGATTCACCAGTGACG	59.1	249	XP_006448817.1
		R	TCCAGTAGACCTGCTGTTTGC	57.9		
28	*ZbWRKY20*	F	ACTGAGGTTCGAGTGGGTGAC	58.6	229	XP_006452514.1
		R	CTGCCTGAGGCTTAAAAAAGG	59		
29	*ZbWRKY47(2)*	F	TCCATTTCCCACCATTACCC	59.4	122	XP_006444621.1
		R	ATGAAGCATGTCGTTGCCTT	58		
30	*ZbWRKY32*	F	ACCAAACAACTTCAACCCAGT	56.2	135	XP_006453624.1
		R	AAGGCTTTATTTCAAACCCGA	58.7		
31	*ZbWRKY44*	F	CACCCAAAACCTCAACCTCCTA	60.8	292	XP_006428919.1
		R	TTCCTTCATCACATTCCCCACT	60.8		
32	*ZbWRKY53(2)*	F	TTTTTAGGGGGTTTCTCCTC	55.5	162	XP_006439376.1
		R	GGCATTTGAATTATTGGCTG	56		
33	*ZbWRKY2*	F	AGACAGCCTCAGCCTCCAAAC	60.8	183	XP_006474751.1
		R	TCCTCAGATGATGCACCTCCA	60.9		
34	*ZbWRKY4(2)*	F	ATCCGCCACCTCAATCTAA	55.2	201	KDO83046.1
		R	CAGCATCACCCCCTTCTTC	57.4		
35	*ZbWRKY1*	F	GCTGTTGGAATTGTCGTGTCT	57.5	81	XP_006447745.1
		R	ATCGGATGATTTGTCTTTGGC	59.1		
36	*ZbWRKY41(2)*	F	AACGAAGAAACGAAATACAACA	55	234	XP_006490540.1
		R	CCTATCAGAAGGACAAACAACA	54.7		
37	*ZbWRKY51*	F	ATCTTGAAGTAATGGATGATGGA	55.8	73	XP_006464492.1
		R	TTATTTGGGTTGTTCTTGACTGA	57.1		
38	*ZbWRKY14(2)*	F	GCAAGGAAACAAGTGGAACG	58.1	214	KDO76342.1
		R	ACTCATCATCAATGGCGGTC	57.7		
UBQ	Ubiquitin extension protein	F	TCGAAGATGGCCGTACATTG	57.5	122	—
		R	TCCTCTAAGCCTCAGCACCA	59.5		

F: Forward primer; R: Reverse primer.

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
