# Peer review of "Patterns of Drought Response of 38 WRKY Transcription Factors of Zanthoxylum bungeanum Maxim."

_ijms, 2018, doi:10.3390/ijms20010068_

Round 1

Reviewer 1 Report

The present manuscript focuses on WRKY transcription factors in Zanthoxylum. The bioinformatics and experimental data are used to build a putative regulatory network in this plant. The present version is of good quality and clear, but please include Figure numbers in the text as they are not currently cited.
The English spelling should be improved before publication.
some minor comments:
please detail how delta delta Ct were calculated. Which sample is the calibrator? (which should be the 0 value in figure3, and apparently does not correspond to D0).
How was performed the clustering in figure4? please complete the legend and material & method part (clustering algorithm, software...) Were the data scaled?
because the final model includes JA as an important hormone, it may have been scientifically sound to include response of Zanthoxylum to JA with similar kinetics to drought.

Author Response

Comments 1: please detail how delta delta Ct were calculated. Which sample is the calibrator? (which should be the 0 value in figure3, and apparently does not correspond to D0).

[Response] Thank you very much for your positive assessment. We have carefully read your constructive comments and suggestions and made detailed revision according to these comments in our revised manuscript point to point. Some mistakes also have been carefully corrected.

The cycle threshold values (Ct) of all samples were collected by the CFX96 Real-Time PCR Detection System, and gene relative expression was calculated as: relative expression i.e. 2-△△Ct. 2-△△Ct=2^-[(Ctsample- Ctreference gene)-(CtD0-Ctreference gene)]

The relative expression levels of all samples were corrected by the above formula, and the final result was corrected based on the control (D0). In fact, the expression level of D0 in Fig. 4 was 1. In this way, the trend of the target gene at different stages can be judged.

Moreover, the language has also been comprehensively re-edited by a native English speaking professional (editor Sandy) to ensure the quality of our writing.

Comments 2: How was performed the clustering in figure4? please complete the legend and material & method part (clustering algorithm, software...) Were the data scaled? because the final model includes JA as an important hormone, it may have been scientifically sound to include response of Zanthoxylum to JA with similar kinetics to drought.

[Response] The analysis of the heat map was performed using the OmicShare website (http://www.omicshare.com/). The principle is to calculate the relative expression amount of the Ct value generated by RT-qPCR by 2-△△Ct, then reduce the value of each value by log10, and then associate one color for each range of values obtained.

Reviewer 2 Report

1. All sections (Introduction, Results and Discussion) were poorly and superficially written.

2. In phylogenetic analysis of WRKYs, authors had done analysis using genes form ZB and A. thaliana only. They can draw a better picture if they will use more species for phylogenetic analysis.

3. Further, nothing available/written in details in result and discussion sections (2.1 to 3). For example, in section “2.3. Conserved domain analysis of ZbKWRKY gene”, only three lines were written, which is actually part of method section.

4. In results, section “2.5. ZbWRKY protein function connection network analysis” written like discussion, and on the basis in STRING result (in silico method) authors conceal a ‘stress signal path model of WRKY participation” in my opinion this is an overseeing of results.

Author Response

Comments 1: All sections (Introduction, Results and Discussion) were poorly and superficially written.

[Response] Thank you for your time and effort in my manuscript. Your comments will definitely help the improvement of the quality of the manuscript. Based on your valuable suggestions, our manuscript has been revised. We realized that the manuscript had a big problem in the introduction, results, and discussion. After our discussion and research, we have rewritten these parts. The specific modification results are presented in the manuscript.

Comments 2: In phylogenetic analysis of WRKYs, authors had done analysis using genes form ZB and A. Thaliana only. They can draw a better picture if they will use more species for phylogenetic analysis.

[Response] Citrus (Citrus sinensis) and Zanthoxylum bungeanum are plants of the family Rutaceae.To verify the correctness of the WRKY transcription factor, in addition to Arabidopsis, we added the WRKY transcription factor for citrus to test the results of the classification. Please refer to Figure 1 in the manuscript for specific modifications.

Comments 3: Further, nothing available/written in details in result and discussion sections (2.1 to 3). For example, in section “2.3. Conserved domain analysis of ZbKWRKY gene”, only three lines were written, which is actually part of method section.

[Response] According to the experts, we realized that this part of the manuscript is not sufficient. Through the analysis of the experimental results and the collected literature, we have reorganized the results and discussion sections, adding results, especially the content of the discussion, to provide a clearer experiment for the reader.

Comments 4: In results, section “2.5. ZbWRKY protein function connection network analysis” written like discussion, and on the basis in STRING result (in silico method) authors conceal a ‘stress signal path model of WRKY participation” in my opinion this is an overseeing of results.

[Response] Based on your comments, we have re-adjusted this section and re-written the "2.5. ZbWRKY Protein Functional Link Network Analysis" in the discussion. Part of the content was added through data search and analysis.

Reviewer 3 Report

The manuscript by Fei et al, regarding the screening of WRKY TFs in Zanthoxylum bungeanum by bio-informatic analyses and qRT PCR aims to evaluate the involvement of these TFs in drought response. According to the data presented, in the reviewer's opinion the manuscript is not suitable for publication in IJMS.

Main critical points are:

The work is not well described and conclusions are not experimentally supported overall the manuscript appears inaccurate

The English language requires extensive editing .In many points the inappropriate use of the verbs led to misleading phrases.

As described in the title, the work should explain the response of WRKY TFs to drought, but in the introduction there is a very poor mention of the specific involvement of this TFs in drought, instead almost all the intro is centered on the structural characterization of WRKY TFs.

The result section lacks of many important points, there is no reference in the text to  any of the figures presented

The authors should indicate which transcriptome they inferred to select the TFs, in which condition the data set has been realized ,from which tissue and developmental stage . Moreover they selected 38WRKY genes and performed qRT only on 12 of these under drought stress. It’s not clear how these genes has been selected out of the 38, moreover a brief description of the stress imposition should be given.

In qRT.PCR experiment the authors should specify why they used ubiquitine as reference gene, if any stability test in drought has been performed before, report the stable expression along the time course, otherwise more than 3 reference genes should be used. A time curse every  6 hours  is performed , why? Finally a statistical analysis of the data should be reported.

Regarding the network analysis in my opinion, 12 genes all belonging to the same family are too little to allow the speculations made on their putative function as reported in the result section. Putative partners as MAP genes for example should have been analyzed by qRT-PCR and included . Anyway the conclusions drown, are not experimentally supported. 

The methods are not accurate, the experimental design is not clear.

I'm sorry but in the present form the manuscript cannot be accepted.

Author Response

Comments 1: The work is not well described and conclusions are not experimentally supported overall the manuscript appears inaccurate.

[Response] Based on your comments, we discussed and found many problems. Therefore, we have made extensive revisions to the introduction, methods, results and analysis, discussion, and conclusions. The logic and causality in the manuscript were reorganized.

Comments 2: The English language requires extensive editing .In many points the inappropriate use of the verbs led to misleading phrases.

[Response] There are many shortcomings in the manuscript in terms of language. So, the language has also been comprehensively re-edited by a native English speaking professional (editor Sandy) to ensure the quality of our writing.

Comments 3: As described in the title, the work should explain the response of WRKY TFs to drought, but in the introduction there is a very poor mention of the specific involvement of this TFs in drought, instead almost all the intro is centered on the structural characterization of WRKY TFs.

[Response] Thank you very much for your attention and patience to our manuscript. We also thank you very much for your thoughtful suggestions and comments followed. We have supplemented the manuscript and revised the sentence, so as to improve the quality of the article under your guidance. We have made adjustments to the introduction and added a description of the WYKYs functionality. At the same time, the logic and language of this part have been modified.

Comments 4: The result section lacks of many important points, there is no reference in the text to any of the figures presented

[Response] Based on the results of the experiment and the predicted results, we rewrote the results. For details of the modifications, please refer to the manuscript lines 695-716.

Due to our negligence, the chart did not correspond to the manuscript. We reorganized the contents of the manuscript and quoted the chart in the corresponding position.

Comments 5: The authors should indicate which transcriptome they inferred to select the TFs, in which condition the data set has been realized, from which tissue and developmental stage. Moreover they selected 38WRKY genes and performed qRT only on 12 of these under drought stress. It’s not clear how these genes has been selected out of the 38, moreover a brief description of the stress imposition should be given.

[Response] Thank you very much for your suggestions. Since our negligence did not explain the source of ZBWRKY, we have now introduced the information of the transcriptome and the source of ZBWRKY in the “Materials” of the manuscript. From the transcriptome of the ZB skin (Laboratory previously obtained), the ZBWRKY transcription factor was initially obtained based on the annotation information. The transcriptome of the Chinese prickly ash skin provides the ZBWRKY transcription factor only as a data source. Based on this, we screened the collected transcription factors. Primers were designed to analyze their expression patterns in different drought stages of Chinese prickly ash. We performed RT-qPCR experiments on all 38 WRKYs, and their expression patterns were made into heat maps Figure 3. We found 12 transcription factors that responded significantly to drought stress by using 5 times more than the control group. We then described the relative expression levels of these 12 transcription factors at different stages to facilitate visual observation of the trends of these transcription factors.

Comments 6: In qRT.PCR experiment the authors should specify why they used ubiquitine as reference gene, if any stability test in drought has been performed before, report the stable expression along the time course, otherwise more than 3 reference genes should be used. A time curse every 6 hours is performed, why? Finally a statistical analysis of the data should be reported.

[Response] Before doing this experiment, we considered the importance of the reference gene for gene expression research. Therefore, we screened the different tissues of Zanthoxylum bungeanum, different developmental stages of fruit, salt stress, drought stress and cold stress. The results obtained indicate that UBQ is suitable as an reference gene of Zanthoxylum bungeanum. The article has been published as: Expression Stabilities of Ten Candidate Reference Genes for RT-qPCR in Zanthoxylum bungeanum Maxim.

Before doing this experiment, we did a preliminary experiment. It was found that the 3 month seedlings of Zanthoxylum bungeanum showed severe wilting after 48 hours of treatment with 20% PEG6000. We also collected samples at 60 hours and 72 hours and found that these samples showed

severe RNA degradation. In order to describe a complete change in gene expression level, we believe that it is appropriate to collect samples from more than 5 stages, so we selected 7 stages from the beginning of 0h for 48 hours, and finally obtained the conclusions in the manuscript through experimental analysis.

Comments 7: Regarding the network analysis in my opinion, 12 genes all belonging to the same family are too little to allow the speculations made on their putative function as reported in the result section. Putative partners as MAP genes for example should have been analyzed by qRT-PCR and included. Anyway the conclusions drown, are not experimentally supported.

[Response] We performed RT-qPCR assay on the identified 38 WRKYs. The relative expression level data of each gene was calculated by the formula, and then the expression trend of the gene was observed by a line graph. The final screening resulted in 12 WRKYs that responded significantly under drought stress. We believe that these genes are involved in the drought of pepper and believe that they may work together with other proteins to function. So we made protein interaction predictions for these corresponding genes. We attempted to construct a regulatory pattern under drought stress in plants that are primarily involved in ZbWRKYs.

Comments 8: The methods are not accurate, the experimental design is not clear.

[Response] We describe the experimental method in more detail, introduce the source of the transcription factor sequence, and the methods of data processing and some websites, trying to clearly show the design and method of the experiment.

Round 2

Reviewer 1 Report

The manuscript has been improved accordind to the reviewers suggestions. In the present form it can be suitable for publication.

Author Response

Thank you for your energy in reviewing the manuscript, and thank you for your valuable suggestions. The quality of the manuscript will be improved under your guidance. After receiving your comments, we discussed and analyzed the problems in our manuscript. The specific results are as follows:

We include our detailed responses to the reviewers. Please note that the comments from the reviewers are in italics followed by our responses in blue-inked text.

Comments 1: line 24-26: "We speculate that WRKY...". The word 'speculate' shows that your research is only based upon speculation, and no evidence whatsoever to support your claim. You should change it into "we predict".

[Response] Based on your comments, we have made changes in the manuscript.

Comments 2:line 93-94: What is the basis of the group division of this gene family? Is it due to the output of the software? or you cite it from gene ontology database?

[Response] The group division of the WRKY family genes is divided according to the conserved domain. For example, the I subfamily has two WRKYGQK regions followed by a C2H2 structure. Each sequence is manually divided according to the WRKY conservative domain structure.

Comments 3:line 110-111: The protein domain conservation is very low. It infers that the functions of the domains are very diverse.

[Response] As you said, the WRKY family of genes involves many regulatory processes, and WRKY family members of the same subfamily may have similar functions, and the functions of members among subfamilies vary widely. It is widely involved in many processes of life regulation.

Comments 4:line 236-252: How did you ilustrate this signal path model? Did you use metabolomics software such as Ingenuity? If you draw it hypothetically without the assistance of any software or computational measure, your predicted pathway is weak.

[Response] This signal path model is constructed based on the interaction between genes. Each pair of interactions is obtained through experiments by predecessors, so each pair is a reliable document.

Comments 5:line 279-280: You should determine the e-value cut-off for the sequence search. It should be from 10 in the power of -3 to -4. You can set it up in the advance search feature of the database. If you use the standard value, there will be imminent false positive in your result.

[Response] It is critical that you ask this question. When we searched the sequence, we realized that if the deviation of the e value is too large, the result will be unreliable. The closer the e value is to 0, the more reliable the sequence. Therefore, when we selected NCBI-BLAST, we strictly controlled the e value and controlled it below 10 in the power of -5.

Comments 6:line 300: Similar question with line 279-280, what is your e-value cut off for your BLAST engine?

[Response] We strictly controlled the e value and controlled it below 10 in the power of -5.

Comments 7:line 333: your conclusion on '30 times expression level' is definitely questionable as you did not specify a more stringent e-value cut off for HMMER and BLAST. False positive could be imminent.

[Response] We initially screened the sequence according to the strict e value, and then designed the primer for RT-qPCR reaction. The obtained data were corrected by the internal reference gene. The obtained result was calculated by the relative expression level of D0, and the relative expression levels of different stages were calculated, to obtain the relative expression level of the target gene at different stages. The relative expression level does not represent the actual expression level, but can indicate a trend of change.

Reviewer 2 Report

The manuscript by Fei et al. describes the screening of abiotic stress related WRKY factors in Zanthoxylum bungeanum. Overall, the manuscript is interesting and appeals to a wide audience. The experiments are well designed and executed. The results are also presented and discussed well. Moreover, the manuscript is written in good English. I do not have any serious issues with the manuscript and therefore, I recommend the article for publication in its current form.

Author Response

Thank you for your energy in reviewing the manuscript, and thank you for your valuable suggestions. The quality of the manuscript will be improved under your guidance. 

Reviewer 3 Report

line 24-26: "We speculate that WRKY...". The word 'speculate' shows that your research is only based upon speculation, and no evidence whatsoever to support your claim. You should change it into "we predict". 

line 93-94: What is the basis of the group division of this gene family? Is it due to the output of the software? or you cite it from gene ontology database?

line 110-111: The protein domain conservation is very low. It infers that the functions of the domains are very diverse. 

line 236-252: How did you ilustrate this signal path model? Did you use metabolomics software such as Ingenuity? If you draw it hypothetically without the assistance of any software or computational measure, your predicted pathway is weak. 

line 279-280: You should determine the e-value cut-off for the sequence search. It should be from 10 in the power of -3 to -4. You can set it up in the advance search feature of the database. If you use the standard value, there will be imminent false positive in your result. 

line 300: Similar question with line 279-280, what is your e-value cut off for your BLAST engine?

line 333: your conclusion on '30 times expression level' is definitely questionable as you did not specify a more stringent e-value cut off for HMMER and BLAST. False positive could be imminent.

Author Response

(The authors gave the same response as above.)

Reviewer 4 Report

ijms-402207

Patterns of drought response of 38 WRKY transcription factors of Zanthoxylum bungeanum Maxim. Xitong Fei et al.

Fei et al. identified 38 ZbWRKY transcription factors and 14 Arabidopsis WRKY by using bioinformatics and classified. They also tried to determine their behaviors under drought stress, and found that twelve ZBWRKY transcription factors were found to be significantly upregulated under drought stress and these were identified by relative quantification. The authors claim the roles of ZbWRKY transcription factors in drought stress rely on a regulatory network center on the JA signaling pathway.

Basically, the data shown in the manuscript are based on bioinformatic analyses and experimental result is only by quantitative RT-PCR. However, the authors have a long discussion section with so many speculative sentences. I think this manuscript do not have enough results and information to support the authors’ conclusion. It is required to perform several experiments and obtain experimental results including nuclear localization, signal transduction pathway activation and DNA binding.

I regret to conclude the manuscript is too primitive to be accepted by IJMS.

Author Response

Thank you for your energy in reviewing the manuscript, and thank you for your valuable suggestions. Thank you very much for your positive assessment. The quality of the manuscript will be improved under your guidance. After receiving your comments, we discussed and analyzed the problems in our manuscript. The specific results are as follows:

We include our detailed responses to the reviewers. Please note that the comments from the reviewers are in italics followed by our responses in blue-inked text.

Comments 1: Fei et al. identified 38 ZbWRKY transcription factors and 14 Arabidopsis WRKY by using bioinformatics and classified. They also tried to determine their behaviors under drought stress, and found that twelve ZBWRKY transcription factors were found to be significantly upregulated under drought stress and these were identified by relative quantification. The authors claim the roles of ZbWRKY transcription factors in drought stress rely on a regulatory network center on the JA signaling pathway.

Basically, the data shown in the manuscript are based on bioinformatic analyses and experimental result is only by quantitative RT-PCR. However, the authors have a long discussion section with so many speculative sentences. I think this manuscript do not have enough results and information to support the authors’ conclusion. It is required to perform several experiments and obtain experimental results including nuclear localization, signal transduction pathway activation and DNA binding.

[Response] Thank you again for your approval of the manuscript. In the manuscript, we obtained the expression of WRKY members in drought mode according to the verification of RT-qPCR, and established a WRKY-centered regulation model based on the interaction between genes. Our role in the WRKY family in a variety of signaling pathways is based on published literature, and each pair of interactions has been fully validated in previous experiments. Therefore, we believe that the established regulatory model is scientifically based.

Round 3

Reviewer 3 Report

After reading the revisions and replies, I recommend to accept this manuscript.

Reviewer 4 Report

I understand the authors think their models are based on the published information. However, they identified ‘new’ WRKY genes. Of course, I understand, the authors assume those factors have similar roles, but nobody knows they have ‘same’ activities. This is why I suggested to obtain experimental data to support their conclusion and model figure. At least, the authors have to show that their RT-PCR primers give them only one PCR product by presenting gel electrophoresis patterns. Even with this, I still think the experimental data are required. However, I would like to yield the right of decision for this point to the editor, since the authors do not agree with me for it.